# Object Detection through Fires Using Violet Illumination Coupled with Deep Learning

**Haojun Zhang** [1], **Xue Dong** [1,*] **and Zhiwei Sun** [2]

1   China-UK Low Carbon College, Shanghai Jiaotong University, Shanghai 201306, China
2   School of Electrical and Mechanical Engineering, The University of Adelaide, Adelaide, SA 5005, Australia
*   Correspondence: xue.dong@sjtu.edu.cn; Tel.: +86-136-364-686-42

**Abstract:** Fire accidents threaten public safety. One of the greatest challenges during fire rescue is that firefighters need to find objects as quickly as possible in an environment with strong flame luminosity and dense smoke. This paper reports an optical method, called violet illumination, coupled with deep learning, to significantly increase the effectiveness in searching for and identifying rescue targets during a fire. With a relatively simple optical system, broadband flame luminosity can be spectrally filtered out from the scattering signal of the object. The application of deep learning algorithms can further and significantly enhance the effectiveness of object search and identification. The work shows that this novel optics–deep learning combined method can improve the object identification accuracy from 7.0% with the naked eye to 83.1%. A processing speed of 10 frames per second can also be achieved on a single CPU. These results indicate that the optical method coupled with machine learning algorithms can potentially be a very useful technique for object searching in fire rescue, especially considering the emergence of low-cost, powerful, compact violet light sources and the rapid development of machine learning methods. Potential designs for practical systems are also discussed.

**Keywords:** fire rescue; violet illumination; deep learning; object detection

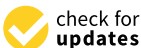



## 1. Introduction

Globally, fires cause over 300,000 deaths each year; millions of people suffer from permanent injuries and approximately 95% of deaths are recorded in low- and middle-income countries [1]. Fire safety refers to the prevention of fires, minimizing the spread of a fire and smoke, extinguishing a fire, and the possibility of a swift and safe evacuation [2].

Effective rescue has always been an enormous challenge to firefighters, as a clear view is essential for firefighters to detect people trapped in fire. However, the radiation from flames is much stronger than that reflected by the objects behind or within the fire, and the existence of soot and smoke further scatters light in random directions, both of which result in a sharp reduction in the signal-to-noise ratio for object detection. To address this issue and improve visibility, it is important to block particularly strong visible and near-infrared radiation from the flames. For example, Hoehler et al. used blue laser triangulation sensors to measure target displacement behind flames fueled by natural gas [3]. In addition, they used matched optical filters in conjunction with narrow-band blue illumination to reduce the influence of optical emissions from a glowing hot target and a large natural gas diffusion flame [4]. Gatien et al. applied narrow-band light with a peak wavelength of 450 nm and a half width of 20 nm, and a digital camera with a frequency-matched optical filter, to capture images of surface charring [5]. These previous works demonstrate that violet illumination is a promising technique for the visualization of targets through a fire. However, there remains a need to further increase the visualization quality, efficiency, and ease of implementation, particularly considering that fire rescue requires a fast response from the firefighters, who may need to work in the field for hours or even longer.

In addition to illumination within a short wavelength, novel image processing methods have also been used to mitigate the influence of flame soot and smoke. Deblurring or dehazing algorithms are typically used to process images. For example, Debnath et al. [6] reported a significant improvement in the contrast of the measured object in the presence of flames and smoke, by implementing the quadrature lock-in discrimination (QLD) algorithm on the images obtained from modulated blue light illumination and imaging. Traditional dehazing methods include those based on image enhancement and those based on physical models [7,8]. The former is achieved by enhancing the edge, contour, and contrast of images, such as Dark Channel Prior (DCP) [8], which means that, on the non-sky area of an RGB hazy-free outdoor image, at least one channel has very low intensity at some pixels. The methods based on physical models aim to study the scattering effect of suspended particles in the atmosphere and retrieve the image by removing scattering. Non-Local Image Dehazing (NLD) [7] is one of these models, which assumes that the colors of a haze-free image can be accurately approximated by hundreds of different colors that form close clusters in the RGB space. Deep learning has also been used extensively in this area in recent years [9–11]. There are two major branches of deep learning regarding dehazing methods: (1) those using convolutional neural networks (CNNs) to generate the parameters of the atmospheric scattering model and restore the dehazed images; and (2) those using an end-to-end CNN (e.g., generative adversarial networks) to generate clear images directly from foggy images [12]. Widely used deep learning dehazing methods include AOD-Net [9], IPUDN [10], and GCANet [11]. AOD-Net, i.e., the All-in-One Dehazing Network, is a convolutional neural network based on a reformulated atmospheric scattering model. IPUDN, i.e., the Iterative Prior Updated Dehazing Network, is a multi-network dehazing framework that utilizes a unique iterative mechanism to estimate and update the haze parameters, transmission map, and atmospheric light. GCANet, i.e., the Gated Context Aggregation Network, is an end-to-end gated context aggregation network for image dehazing and deraining.

Moreover, target detection through fires is also of great importance for fire rescue. This requires the application of object classification and detection algorithms, which is one of the most fundamental methods in the area of computer vision. It has been applied to image classification, human behavior analysis, face recognition, etc. [13–18]. In general, object detection algorithms can be divided into two categories. One is a two-stage detector, such as Regions with Convolutional Neural Network features (R-CNN) [19], Faster R-CNN [20], and Mask R-CNN [21]. The other concerns single-stage detectors such as You Only Look Once (YOLO) [22], Single-Shot Multi-Box Detector (SSD) [23], and RetinaNet [24]. The two-stage detector has higher target localization and recognition accuracy, while a one-stage detector has a higher inference speed. Bianco et al. proposed a method combining an infrared active imaging sensor and Mask R-CNN to achieve the automatic detection of people obscured by flames [25]. As a representative image object detector, YOLO is often selected to detect humans. The minimum average accuracy of YOLO is more than twice that of other real-time object detection methods, and the background errors of YOLO are smaller than those of Fast R-CNN. For example, Marina et al. reported human detection on a customized dataset of thermal videos using the out-of-the-box YOLO convolutional neural network [26]; however, the accuracy and detection speed in this work were insufficient for real-time fire rescue. In the current research, an updated version (YOLOv5) is therefore adopted to detect objects, also considering that YOLOv5 outperforms other methods for real-time object detection [27,28].

In this work, we propose a method combining narrow-spectrum violet illumination and imaging, deep-learning-based dehazing, and object detection algorithms to improve the visibility of targets behind or within a fire. The workflow of the current research is shown in Figure 1. The main contents of this work include the following:

(1)	The use of a 405 nm LED light source, a CMOS camera, and a matched band-pass optical filter to capture images of targets under different conditions of flames, thereby reducing the obstruction caused by flames and enhancing the signal-to-noise ratio;

(2)   The application of a dehazing algorithm in image processing; several dehazing algorithms are used to ameliorate the blocking effect of smoke and soot;

(3)   The application of the YOLOv5 object detection algorithm to detect the targets behind flames and to improve the detection accuracy by training the deep learning model with images collected from fire scenes.

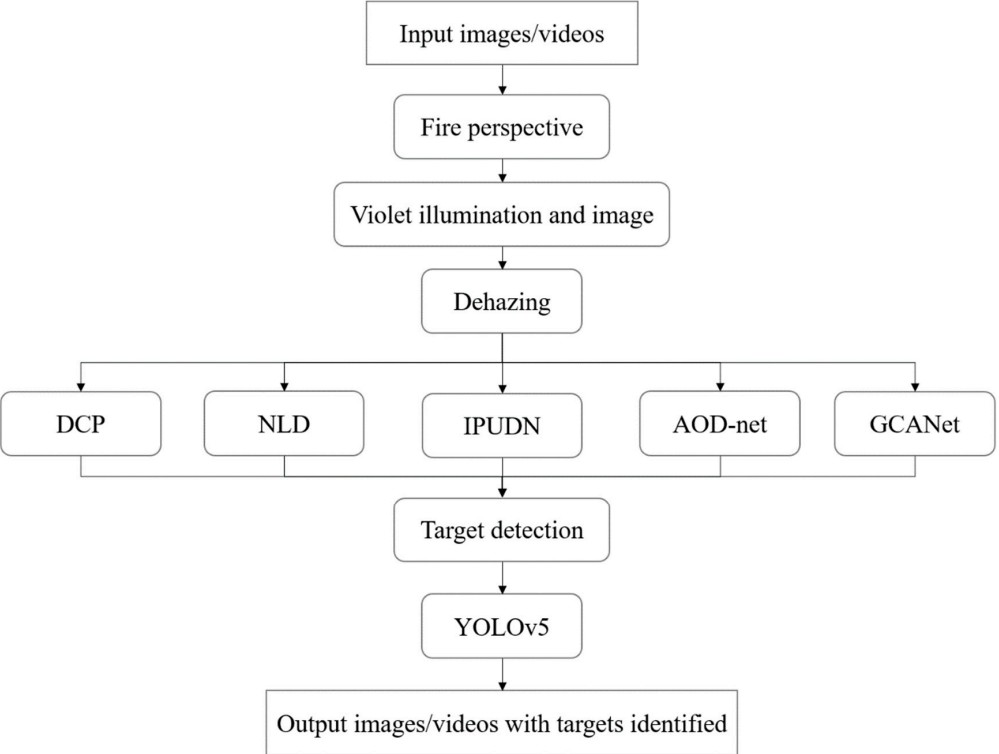

**Figure 1.** Workflow diagram of flame perspective object detection algorithm.

This work proposes a relatively simple (in optics) yet effective violet illumination and imaging (VII) approach to visualize targets through a fire, and also introduces a dehazing and object detection neural network to achieve the real-time detection of targets through a fire. Both of these approaches have great potential to benefit fire rescue.

## 2. Methodology

### 2.1. Experimental Setup

The experiments were conducted outdoors on sooty flames stabilized on a burner fueled with solid materials (wood, paper, cotton clothes, and their combinations, essentially covering common indoor fire sources in an actual fire disaster situation). As shown in Figure 2, for the violet illumination, a high-power LED light source (model CEL-LED100HA) was used, with a wavelength of 405 nm and a full width at half maximum of 20 nm. The spectrum of the violet illumination is shown in Figure 3. The outlet diameter of the LED beam was 50 mm, with a total input power of 240 W (±0.01%) and an optical output power of 20 W (±1%). The LED emitting light had a divergence angle of 24°, rather than being collimated. For imaging, a CMOS camera with a resolution of 2040 × 1086 pixels (model acA2000-165uc, manufactured by Basler) was employed. Moreover, a band-pass filter of 405 nm was applied in front of the camera to reduce the interference from the flame illumination while maximizing the signal from the reflected LED illumination.

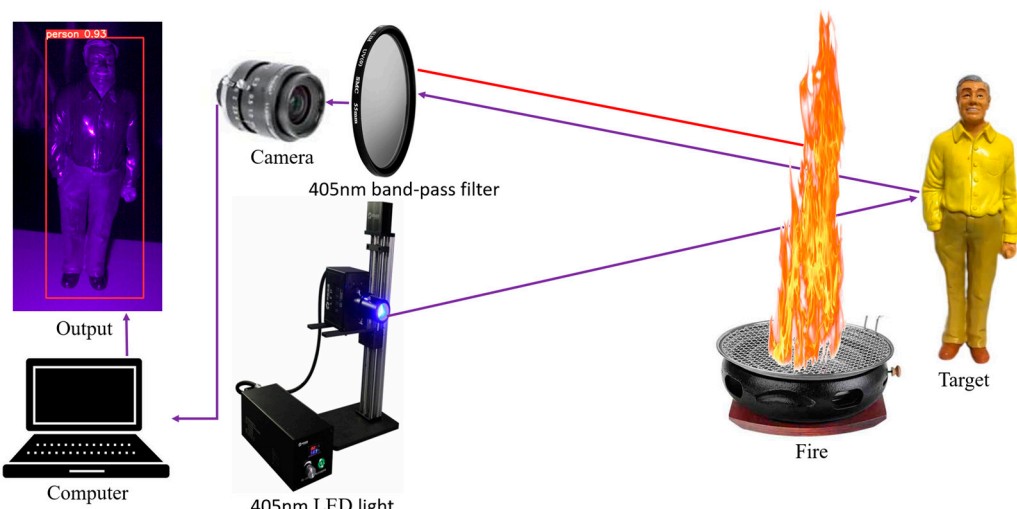

**Figure 2.** Schematic diagram of the experimental setup.

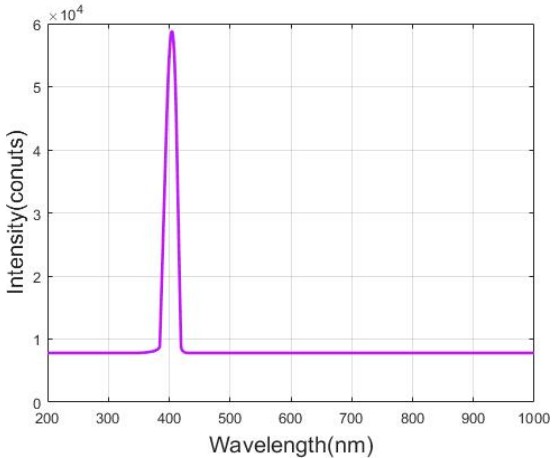

**Figure 3.** Spectrum of the violet LED light.

A sequence of images was taken to cover various transient states of flames. Two human miniature models of 10 cm in height were located behind the flames as targets, and the light and camera were placed at roughly the same height as the target. The distance from the target to the flame was 30 cm, that from the flame to the LED light was 50 cm, and that from the LED light to the camera was 10 cm. During illumination and imaging, we ensured that the flame completely covered the target. A total of 3975 images were acquired, which included 584 naked eye images (without violet illumination and band-pass filtering) and 3401 VII images (2348 images were covered by flames and 1053 images were not covered by flames). It is noteworthy that the images without violet illumination were taken as a control group; hence, images with and without violet illumination were not taken at the same moment. However, as was mentioned previously, the flames were always sufficient to cover the target (human model) behind the flames; hence, the comparison experiments did not need to be conducted at the same time.

### 2.2. Haze Removal Methods

As the occlusion of soot and smoke is similar to that of haze in violet illumination and imaging, various dehazing methods are evaluated in this study, which are briefly explained below.

### 2.2.1. Dark Channel Prior

The atmospheric scattering model is a classical description used to generate images of foggy days. This model is expressed by the following equation:

$$I(x) = J(x)t + A(1 - (x)) \tag{1}$$

where I(x) is the observed intensity, J(x) is the scene radiance to be recovered, A is the global atmospheric light, and t(x) is the transmission matrix defined as

$$t = e^{-\beta d(x)} \tag{2}$$

where β is the scattering coefficient of the atmosphere, and d(x) is the distance between the object and the camera.

Dark channel prior means that on the non-sky area of an RGB hazy-free outdoor image, at least one channel has very low intensity at some pixels. Thus, an image J is defined as

$$J^{dark}(x) = \min_{c \in \{r,g,b\}} \left( \min_{y \in \Omega(x)} (J^c(y)) \right) \tag{3}$$

where $J^c(y)$ is a color channel of J and $\Omega(x)$ is a local patch at x.

### 2.2.2. Non-Local Image Dehazing

Non-Local Image Dehazing is an algorithm based on a novel non-local prior. The clusters are usually non-local, meaning that their pixels are distributed across the entire image plane and are located at different distances from the camera. When haze is present, these varying distances are converted into different transmission coefficients. Consequently, each color cluster in the clear image becomes a line in the RGB space of the hazy image. This algorithm can restore the distance map and haze-free image by applying these fog lines without requiring any training.

### 2.2.3. AOD-Net

AOD-Net generates a clean image through a lightweight CNN, instead of estimating the transmission matrix and the atmospheric light separately. Most dehazing work is based on estimating t(x) and A individually, but this can lead to the accumulation of estimation errors. To address this issue, AOD-Net aims to estimate both parameters in a unified manner K(x) with the following reformulation of Equation (4):

$$K(x) = \frac{\frac{1}{t(x)}(I(x) - A) + (A - b)}{I(x) - 1} \tag{4}$$

where A and t(x) are integrated into a single variable K(x), which depends on the input I(x).

### 2.2.4. IPUDN

IPUDN utilizes a unique iterative mechanism to estimate and update the haze parameters, transmission map, and atmospheric light. Through specific convolutional networks, color cast processing is enabled for the initial estimation of these parameters. These estimates are then used as priors in the dehazing module, where they are iteratively updated through new convolutional networks. A joint updating and dehazing process is carried out by a convolutional network that invokes inter-iteration dependencies, allowing for the gradual modification of the haze parameter estimates to achieve optimal dehazing.

### 2.2.5. GCANet

GCANet restores the clean image directly, without relying on traditional image priors such as dark channels and increased contrast. This network utilizes a smoothed dilation convolution and gated sub-network to eliminate the gridding effect and fuse features from different levels. The overall network structure of GCANet consists of three convolution

blocks in the encoder part and one deconvolution block with two convolution blocks in the decoder part, as illustrated in Figure 4. Smoothed dilated resblocks are inserted between these two parts to aggregate context information without introducing gridding artifacts. The features from different levels are then fused by a gated fusion sub-network for improved image restoration.

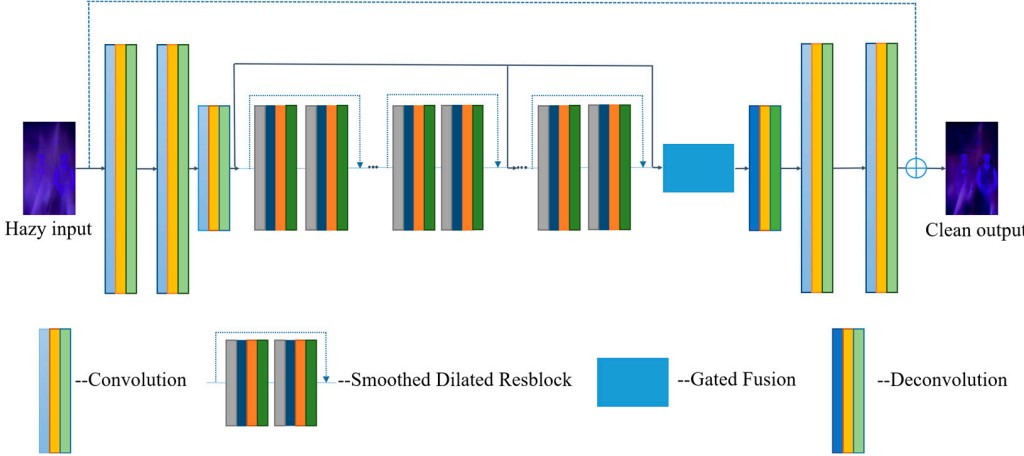

**Figure 4.** The network structure of GCANet.

In this method, the ground truth is defined as

$$r(x) = J(x) - I(x) \tag{5}$$

The predicted haze residue is defined as

$$\hat{r}(x) = GCANet(I(x)) \tag{6}$$

The loss function is defined as the mean square error loss:

$$L = ||\hat{r}(x) - r(x)||^2 \tag{7}$$

### 2.3. Evaluation Indices

To quantitatively evaluate the similarity between two images, the Structure Similarity (SSIM) Index and Peak Signal to Noise Ratio (PSNR) were employed in the research. The SSIM of the two images, x and y, can be calculated using the following equation:

$$SSIM(x, y) = \frac{\left(2\mu_x\mu_y + C_1\right)\left(2\sigma_{xy} + C_2\right)}{\left(\mu_x^2 + \mu_y^2 + C_1\right)\left(\sigma_x^2 + \sigma_y^2 + C_2\right)} \tag{8}$$

where $\mu_x$ is the mean value of x, $\mu_y$ is the mean value of y, $\sigma_x^2$ is the variance of x, $\sigma_y^2$ is the variance of y, and $\sigma_{xy}$ is the covariance of x and y. $C_1 = (0.01\ L)^2 = 6.5025$ and $C_2 = (0.03\ L)^2$ are both constants, where L = 255 represents the image gray level. The SSIM is calculated by a 11 × 11 pixel sub-window and averaged to the full scale of the image to form a global value.

PSNR is a widely used index to compare the similarity of two images pixel by pixel. It is calculated as the ratio of the maximum possible power of a signal to the power of the noise and is expressed as follows:

$$PSNR = 10 \times \log_{10}\left(\frac{MAX^2}{MSE}\right) \tag{9}$$

where MAX = 255 and the mean square error (MSE) of two images x and y can be computed as

$$MSE = \frac{1}{m \times n} \sum_{i=1}^{m \times n} (x - y)^2 \tag{10}$$

## 3. Object Detection Algorithms

The YOLOv5 pre-trained model is based on the Common Objects in Context (COCO) dataset, which contains more than 330,000 images (200,000 labeled) with 80 different classes (including the class person). Therefore, YOLOv5 is suitable and convenient for human detection. We used 1053 images during model training, with the number of images in the training set, validation set, and test set being 842, 106, and 105, respectively. The training was conducted on an NVIDIA GeForce RTX 2080 Ti GPU, while a total of 5.387 h was required to train 250 epochs with a batch size of 16.

## 4. Results and Discussion

The representative results of the original flame images, violet illumination and imaging (VII), and deep learning algorithms for dehazing and target detection through fire are shown in Figure 5. It can be seen that people hidden behind the flames are completely invisible to the naked eye. After using 405 nm LED light and a matched band-pass optical filter, the outlines of the people could be roughly seen. Then, different dehazing algorithms were applied to the VII images to select the optimal algorithm, and the YOLOv5 object detection algorithm trained with a self-made dataset was used to identify the target.

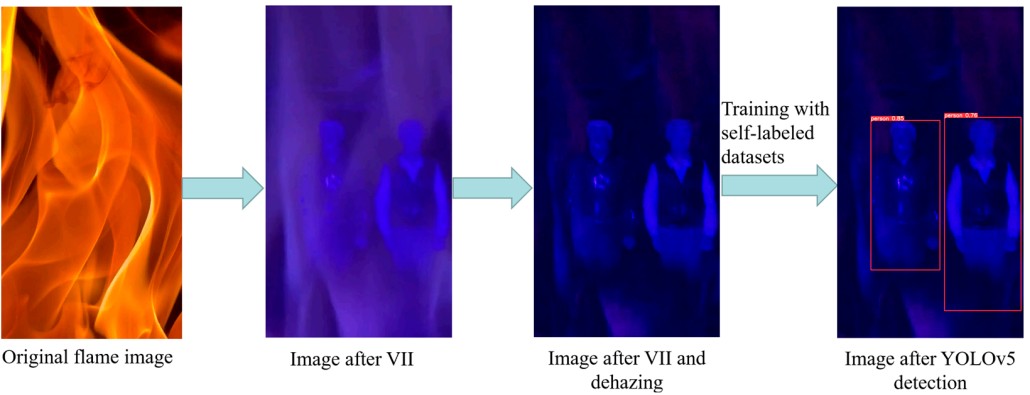

Original flame image　　Image after VII　　Image after VII and dehazing　　Image after YOLOv5 detection

**Figure 5.** Representative results of original flame images, violet illumination and imaging (VII), and deep learning algorithms for dehazing and object detection.

The qualitative and quantitative comparisons of the violet light illumination and imaging (VII) images under different haze removal methods are shown in Figure 6. VII without flames was applied as the ground truth. Compared to the VII image (second column), each method improved the quality of the image. It is also shown that the image evaluation indices (SSIM and PSNR) of a single image were increased when employing the dehazing algorithms. Specifically, SSIM was increased by 0.014, 0.0021, 0.1642, 0.1552, and 0.1745 under Non-Local Image Dehazing, Dark Channel Prior, AOD-Net, IPUDN, and GCANet, respectively, and the PSNR of each method increased by 1.1561, 0.1638, 3.079, 2.7089, and 3.2799, respectively. The statistical results of SSIM and PSNR for 427 images are shown in Figure 7. The statistical SSIM and PSNR of images processed by dehazing algorithms are distributed around higher indices, and GCANet has the best results among several dehazing algorithms.

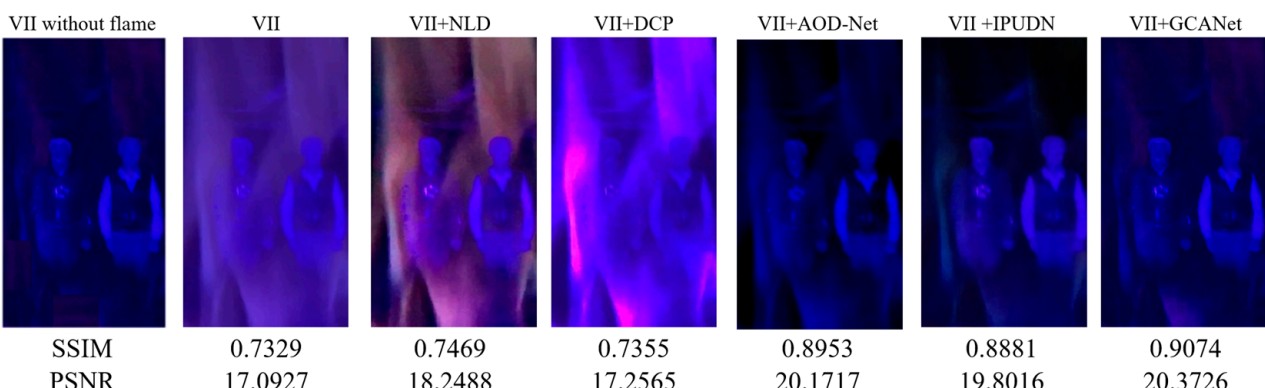

| | VII | VII+NLD | VII+DCP | VII+AOD-Net | VII +IPUDN | VII+GCANet |
|---|---|---|---|---|---|---|
| SSIM | 0.7329 | 0.7469 | 0.7355 | 0.8953 | 0.8881 | 0.9074 |
| PSNR | 17.0927 | 18.2488 | 17.2565 | 20.1717 | 19.8016 | 20.3726 |

**Figure 6.** Comparisons of the images taken with violet illumination and processed with different haze removal methods.

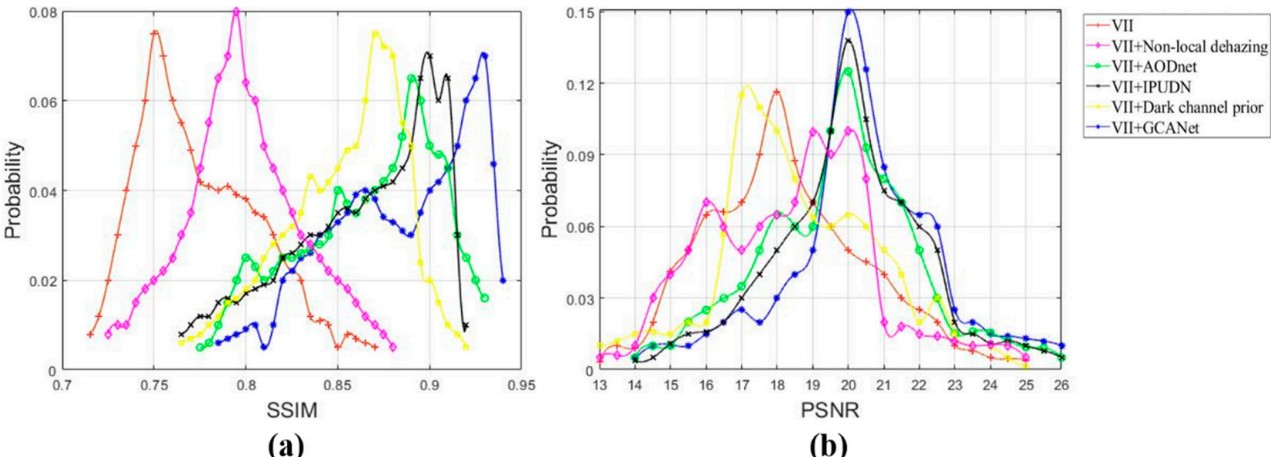

**Figure 7.** Statistical results of SSIM and PSNR in the captured image dataset. (**a**) SSIM; (**b**) PSNR.

In Figure 8, we also present snapshots obtained with and without object detection algorithms for the original fire images, the images with VII, and the images with combined VII and GCANet. It can be seen from Figure 8a that the people hidden behind the flames were blocked when the images were taken without VII. After using VII, the outlines of the people, shown in Figure 8b, could be roughly seen. However, the visibility was not sufficient due to occlusion from soot and smoke. As is shown in Figure 8c, the visibility improved significantly and the figure could be clearly seen after the image was processed by GCANet. Figure 8d–f show the results of YOLOv5 detection. We can see that no target could be identified when the images were shot without any LED light or filter. After using 405 nm LED light and a matched band-pass optical filter, a few targets could be identified. After images were processed by GCANet, the accuracy and probability of identification were both improved by a large margin.

Here, the detection rate is defined as the ratio of the number of correctly recognized images to the total number of images (1053 images), under an illumination distance of 0.5 m. The comparison of the YOLO detection rate on the self-trained model and pre-trained model is presented in Table 1. As can be seen from the table, when using YOLOv5s pre-training weights, VII alone can improve the detection rate from 7.04% to 30.4%. Moreover, different dehazing algorithms can further increase the detection rate, with the best performance of 49.7% achieved by GCANet. Although the detection rate of the best dehazing algorithm is improved more than seven times compared to the original image, there is still room for improvement after using self-trained weights, with the detection rate increased to 83.1%, which is more than 10 times higher than that of the original flame images.

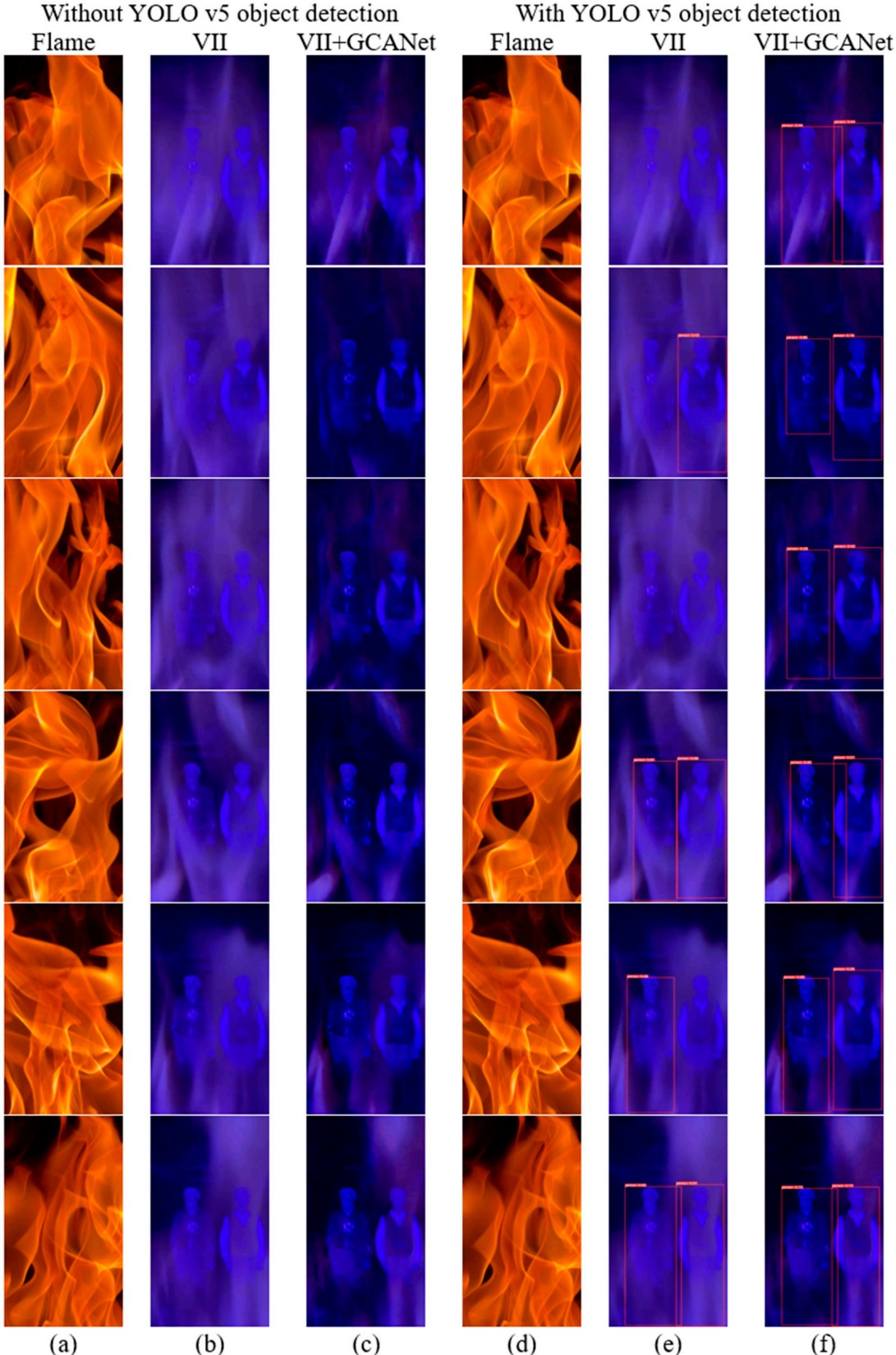

**Figure 8.** Comparison before and after YOLOv5 detection. (**a**) Naked eye flame images; (**b**,**c**) VII images before and after being processed by GCANet; (**d**–**f**) YOLO detection results of naked eye flame image and VII images before and after processing with GCANet.

**Table 1.** Comparison of YOLO detection rate by pre-trained YOLO model and self-trained model, under different dehazing methods.

| Detection Rate | Flame Image | VII | VII + NLD | VII + DCP | VII + AODNet | VII + IPUDN | VII + GCANet |
|---|---|---|---|---|---|---|---|
| Pre-trained YOLOv5s model (%) | 7.04 | 30.4 | 32.8 | 33.1 | 46.3 | 48.0 | 49.7 |
| Self-trained model (%) | 2.11 | 50.6 | 53.1 | 44.4 | 72.5 | 77.5 | 83.1 |

It is worth pointing out that most of the current state-of-the-art (SOTA) methods for fire target detection are based on naked eye flame images, while the flame images obtained based on the narrow-band violet illumination and imaging method described in this paper have not been used in previous SOTA fire detection methods, so the effectiveness of this method in fire detection benefits from combined effects of VII, the dehazing algorithm, and the object detection algorithm.

To verify the effect on the detection rate of VII, the dehazing algorithm, and the self-trained YOLOv5 object detection algorithm module, we also conducted ablation experiments. Table 2 shows the results of the ablation experiments with different modules. The results show that the detection rate can be improved to 30.4%, 49.7%, and 50.6% by VII, VII + dehazing algorithm, and VII + self-trained YOLOv5 object detection algorithm, respectively. Combining VII, the dehazing algorithm, and the self-trained YOLOv5 object detection algorithm can improve the detection rate to 83.1%.

**Table 2.** Comparison of ablation of different modules.

| Original Flame | VII | Dehazing Algorithm | Self-Trained YOLOv5 | Detection Rate (%) |
|---|---|---|---|---|
| √ | | | | 7.04 |
| √ | √ | | | 30.4 |
| √ | √ | √ | | 49.7 |
| √ | √ | | √ | 50.6 |
| √ | √ | √ | √ | 83.1 |

Table 3 compares the processing times of different dehazing algorithms combined with the YOLOv5 algorithm for a single image. Since the processing time of YOLOv5 is only 0.007 s, the overall processing time mainly depends on the efficiency of the dehazing algorithm. The inference time of AODNet + YOLOv5 and GCANet + YOLOv5 for a single image is only 0.116 s and 0.101 s, respectively, while the processing time of other dehazing algorithms is longer than 1 s, making them unable achieve real-time performance. It is worth pointing out that an Intel i7-9700K CPU was used for algorithm deployment and image processing in this study, and the processing is expected to be further accelerated if a GPU is used.

**Table 3.** Comparison of processing time for single image using different dehazing algorithms combined with YOLOv5.

| Dehazing Algorithm | NLD + YOLOv5 | DCP + YOLOv5 | AODNet + YOLOv5 | IPUDN + YOLOv5 | GCANet + YOLOv5 |
|---|---|---|---|---|---|
| Processing time (s) | 3.964 | 2.047 | 0.116 | 1.275 | 0.101 |

The convergence of the self-trained YOLOv5 model is shown in Figure 9. Box loss indicates how well the algorithm can locate the center of an object and how well the predicted bounding box covers an object. The box loss function is defined as

$$l_{box} = \lambda_{coord} \sum_{i=0}^{S^2} \sum_{j=0}^{B} I_{i,j}^{obj} bj \left(2 - w_i \times h_i\right) \left[ \left(x_i - x\wedge_i^j\right)^2 + \left(y_i - y\wedge_i^j\right)^2 + \left(w_i - w\wedge_i^j\right)^2 + \left(h_i - h\wedge_i^j\right)^2 \right] \quad (11)$$

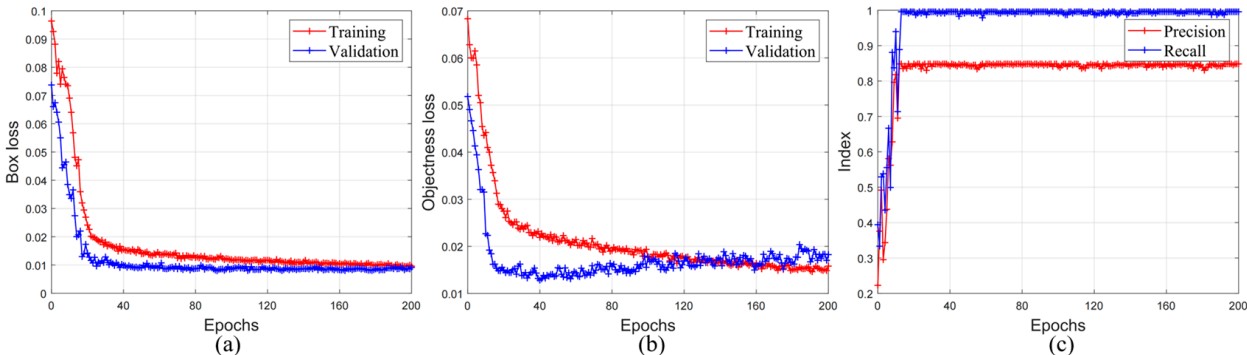

**Figure 9.** The training results of YOLOv5. (**a**) Box loss; (**b**) objectness loss; and (**c**) precision and recall.

Objectness loss measures the probability that an object exists in a region of interest. The objectness loss function is defined as

$$l_{obj} = \lambda_{noobj} \sum_{i=0}^{S^2} \sum_{j=0}^{B} I_{i,j}^{noobj}(c_i - c\wedge_l)^2 + \lambda_{obj} \sum_{i=0}^{s^2} \sum_{j=0}^{B} I_{i,j}^{obj}(c_i - c\wedge_l)^2 \qquad (12)$$

where $\lambda_{coord}$ is the position loss coefficient; $x\wedge_i^j$, $y\wedge_i^j$ are the true central coordinates of the target, and $w\wedge_i^j$, $h\wedge_i^j$ are the width and height of the target. If the anchor box at (i, j) contains targets, then the value $I_{i,j}^{obj}$ is 1; otherwise, the value is 0.

After 200 epochs, the training and validation losses appeared to be stable, indicating that the network converged. The model was also improved in terms of precision and recall. Furthermore, we evaluated the sensitivity of the detection rate to the confidence threshold of YOLOv5. Figure 10 presents the comparison of the detection rates of images processed with GCANet under different confidence thresholds with YOLOv5 self-trained weights. The default confidence threshold of YOLOv5 is 0.25. With the confidence thresholds set to 0.1, 0.15, 0.2, 0.25, 0.3, and 0.35, the detection rates are 72.6, 76.6, 80.8, 83.1, 81.3, and 79.4, respectively. Decreasing the confidence threshold increases the number of detected target frames, but also increases the probability of false detection. Conversely, increasing the confidence threshold decreases the number of detected target frames. Thus, we chose 0.25 as the optimized confidence threshold of YOLOv5.

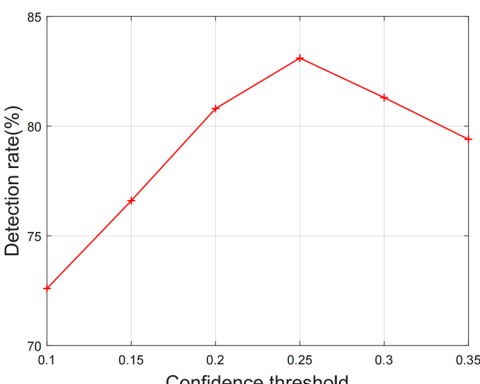

**Figure 10.** Comparison of detection rate under different confidence thresholds.

As the incident light intensity may also influence the performance of this method, in Figure 11, we compare the detection rate of the best dehazing methods under different violet light intensities. It is shown that the recognition rate decreases with the decrease in the violet light intensity. This is because the people behind the flames are not illuminated sufficiently. At the same time, we also studied the influence of the fire wall thickness on the detection rate. It was shown that with the increase in the fire wall thickness, the

detection rate also decreased. The initial fire wall thickness was 10 cm. When the fire wall thickness was increased by 10 cm, the detection rate decreased from 83.1% to 50.4%. It should be noted that the local violet flux of the object can be further enhanced by reducing the divergence angle of the light while maintaining the optical output of the LED.

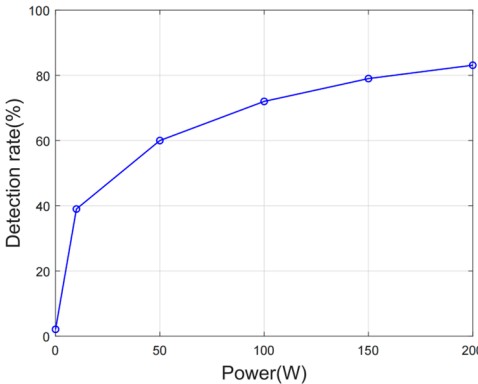

**Figure 11.** Comparison of detection rates of the best dehazing methods under different violet LED light intensities.

To study the impact of the light source distance on the object detection results, we also replaced the light source and camera for the experiments. The beam angle of the light source was 25°. The input power of the light source was 30 W, and the output power was 2500 mW. The test set included a total of 1921 images, and the experimental results were still satisfactory in the test set. Table 4 shows that when the illumination distance increased from 3 m to 15 m, the targets were less clearly illuminated and the detection rate gradually decreased from 80.8% to 40.4%. The size of the dataset collected at each illumination distance was around 385.

**Table 4.** Comparison of YOLOv5 object detection results at different light source distances.

| Illumination Distance (m) | 3 | 6 | 9 | 12 | 15 |
|---|---|---|---|---|---|
| VII + GCANet + YOLOv5 | 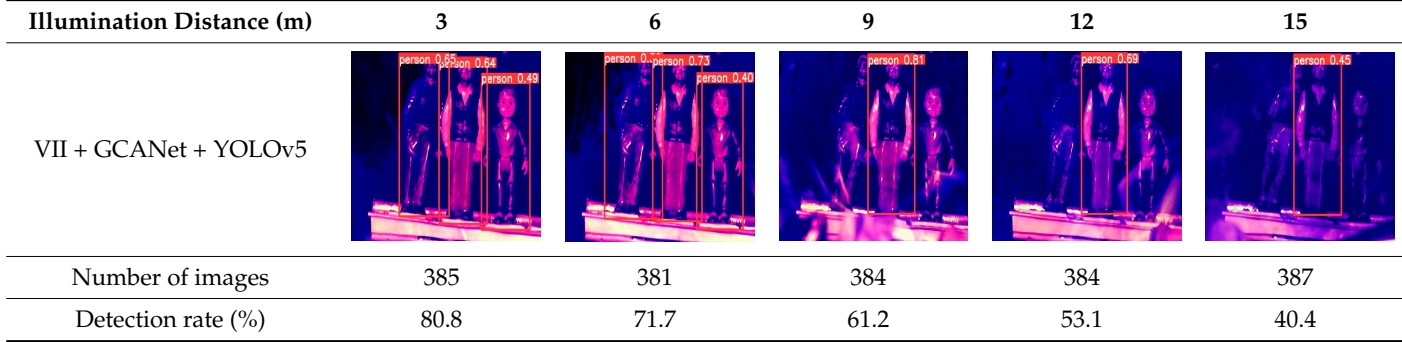 | | | | |
| Number of images | 385 | 381 | 384 | 384 | 387 |
| Detection rate (%) | 80.8 | 71.7 | 61.2 | 53.1 | 40.4 |

## 5. Conclusions

A method based on violet illumination and imaging, the deep-learning-based dehazing and object detection algorithm, was introduced to detect targets behind a fire. To mitigate the interference of the luminosity of flames, a system integrating a 405 nm LED light source, a matched band-pass filter, and a CMOS camera was designed. Several dehazing algorithms, including Dark Channel Prior, Non-Local Image Dehazing, AOD-Net, IPUDN, and GCANet, were applied to further reduce the influence of soot and smoke, and thus to improve the image quality. The results on single and multiple images showed that the detection accuracy for targets behind flames was significantly improved from 7.04% to 30.4% by violet illumination alone. Applying different dehazing algorithms to VII images further increased the detection accuracy, with the best performance of 49% achieved by GCANet. Moreover, the self-trained YOLOv5 model could further increase the detection accuracy to 83.1%. In addition, the inference time of the dehazing algorithm and object

detection algorithm was 0.101 s on a CPU that could achieve a processing speed of 10 FPS. This work demonstrates a relatively simple (in optics) approach for object detection through fire, which could potentially benefit fire rescue.

**Author Contributions:** Conceptualization, X.D. and Z.S.; methodology, X.D. and H.Z.; software, H.Z.; validation, H.Z.; formal analysis, H.Z.; investigation, X.D. and H.Z.; resources, X.D. and H.Z.; data curation, H.Z.; writing—original draft preparation, H.Z.; writing—review and editing, X.D. and Z.S.; visualization, H.Z.; supervision, X.D. and Z.S.; project administration, X.D. and H.Z.; funding acquisition, X.D. All authors have read and agreed to the published version of the manuscript.

**Funding:** This research was funded by the National Natural Science Foundation of China under Grant No. 52006137 and the Shanghai Sailing Program under Grant No. 19YF1423400.

**Institutional Review Board Statement:** Not applicable.

**Informed Consent Statement:** Not applicable.

**Data Availability Statement:** The data presented in this study are available on request from the corresponding author.

**Conflicts of Interest:** The authors declare no conflict of interest.

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
