# Peer review of "Object Detection through Fires Using Violet Illumination Coupled with Deep Learning"

_fire, doi:10.3390/fire6060222_

Round 1

Reviewer 1 Report

Summary:

The paper provides an approach for identifying and detecting human targets behind flame and improving detection accuracy by using a deep learning-based computer vision model.

Comment:

  1. The authors have the objectives of their work in the paper but it is unclear what are the key contributions and novelties of the work. Please add contribution statements to the paper.
  2. Add the workflow diagram to the paper and discuss it in the introduction section to describe the complete approach. This readers to clearly understand the steps undertaken in the study.
  3. The authors have claimed on line 100 that there are insufficiencies in previous work but, the study did not back the claim by conducting any comparison with pre-existing approaches (ex. SOTA). The comparison will further strengthen the impact of the study.
  4. The number of samples in the training data is small. While trained computer vision models are used in the study it is unclear how the models developed in the study will be useful in practice. As in real evacuation, a multitude of flame scenarios may arise.
  5. In an actual fire situation, typically, the humans will be moving, and the objects in the buildings would have moved or fallen apart. How will the model work in uncertain situations? A more thorough and concrete analysis is required to consolidate the paper. 
  6. It is unclear what are the typical limitations of the work. 
  7. Figure(8)What is "Box Loss " and "objectiveness Loss" mathematical definition? Please include it for readers' clarity.
  8. Improve the language in the paper. Use "present perfect tenses" in sentences as much as possible. 

Author Response

We appreciate the reviewer’s comments, please find our detailed response as attached!

Reviewer 2 Report

The information in article solve problem of visibility in smoke and the presence of bright flames. This paper proposes a new optical method called violet illumination and imaging to eliminate interferences from flame luminosity. This is an important problem during fire rescue operations, especially in buildings. I have not encountered similarly presented analyzes in the literature. The results have an important practical aspect.

The article is interesting but:

1. The conclusions are presented in the form of a summary. It is worth supplementing it with specific statements supported by numbers and a practical aspect.

2. The literature review consists of 42 literature items. It covers various areas of this topic. The studies come from different regions of the world. I evaluate it good. 

3. A discussion of the results should be added after the information presented in the figures. This is a clear lack in the article.

Author Response

(The authors gave the same response as above.)

Round 2

Reviewer 1 Report

Thank you for taking the time to address the comments provided on the revised manuscript. However, I have noticed that some of the comments have not been fully addressed. Please find below the remaining comments: 1. Although the text regarding the steps has been added, it would be beneficial to include a workflow diagram to provide a comprehensive understanding of the process. 2. The contribution statements lack novelty and do not meet the standards set by the journal. 3. While the authors claim to cover different scenarios for flame generation in the reviewer response, the design of experiments has not been described in the paper. Additionally, the dataset samples exhibit heterogeneity, but they are still small. 4. The conclusion in the revised version is too lengthy. I hope you find this feedback helpful in improving the manuscript.

Author Response

We appreciate the reviewer's comments, detailed response can be seen from the attached documents. 

Reviewer 2 Report

The revised article is ready for publication.

Author Response

We appreciate the reviewer's comments, please find the attached documents for detailed response. 
